# Citizens can help to map putative transmission sites for snail-borne diseases

**Julius Tumusiime**[1,2☉]*, **Noelia Valderrama Bhraunxs**[3,4☉], **Grace Kagoro-Rugunda**[1], **Daisy Namirembe**[1], **Christian Albrecht**[1,2], **Ronald Twongyirwe**[5], **Casim Umba Tolo**[1], **Liesbet Jacobs**[4,6], **Tine Huyse**[3]

**1** Department of Biology, Mbarara University of Science and Technology, Mbarara, Uganda, **2** Institute of Animal Ecology and Systematics, Justus Liebig University Giessen, Giessen, Germany, **3** Department of Biology, Royal Museum for Central Africa, Tervuren, Belgium, **4** Department of Earth and Environmental Sciences, KU Leuven, Leuven, Belgium, **5** Department of Environment and Livelihoods Support Systems, Mbarara University of Science and Technology, Mbarara, Uganda, **6** Ecosystem and Landscape Dynamics, Institute for Biodiversity and Ecosystem Dynamics, University of Amsterdam, Amsterdam, The Netherlands

☉ These authors contributed equally to this work.
\* jtumusiime90@must.ac.ug

## Abstract

### Introduction

Schistosomiasis and fasciolosis are snail-borne diseases of great medical and veterinary health importance. The World Health Organization recommends complementing drug treatment with snail control and community involvement for disease elimination, but there is a general lack of snail experts and hence snail distribution data. Therefore, we adopted a citizen science approach and involved citizens in the monitoring of medically and veterinary important snail taxa.

### Materials and methods

Snail data was collected weekly by 25 trained citizen scientists (CSs) at 76 sites around southern Lake Albert (Uganda) for 20 months. At each site, snails were searched for 30 minutes, sorted, target snail hosts identified to genus level, counted and data submitted through a smartphone application. The quality of this data was assessed by comparing it to monthly data collected by an 'expert' malacologist using the same sampling protocol. Generalised binomial logistic and linear mixed-effects models were used to analyse the variables for agreement between the CSs and expert.

### Findings

The binary agreement in presence/absence of *Biomphalaria*, *Bulinus* and *Radix* snails reported by the expert and CSs ranged between 70% and 86% (900 reports) with an average of 17% false negatives (sites wrongly defined as snail-free). The agreement for *Biomphalaria* and *Radix* increased with snail abundance, and false negatives decreased when the number of snails collected by citizens was aggregated per month. Site type significantly predicted binary agreement, which was lowest at lake sites (55%) and highest at spring sites (99%) with variations across genera. Similar temporal trends in snail abundance were

**Data Availability Statement:** The data used in this paper is available in a public repository figshare with the DOI 10.6084/m9.figshare.25047296 at the link https://figshare.com/articles/dataset/

Comparison_data_for_freshwater_snails_hosts_of_bilharzia_and_liver_flukes_collected_by_citizen_scientists_and_an_expert_in_the_Lake_Albert_region_of_Uganda/25047296. The code for validation of the citizen scientist collected is available on GitHub is available at the link https://github.com/CiSciUganda/data_validation/blob/main/Automatic_KOBO.

**Funding:** This research was financed by the Action Towards Reducing Aquatic snail-borne Parasitic diseases (ATRAP) project of the Development Cooperation program of the Royal Museum for Central Africa and Mbarara University of Science and Technology with support of the Directorate-General Development Cooperation and Humanitarian Aid (JT, NVB, CUT, DN, GK, RT, LJ & TH). NVB was also funded by a FWO fellowship grant of the Research Foundation–Flanders (FWO–Vlaanderen 11L3223N). The funders had no role in study design, data collection and analysis, decision to publish, or preparation of the manuscript.

**Competing interests:** The authors have declared that no competing interests exist.

recorded despite the expert reporting higher abundance. However, the relative abundance was consistent across site types. The match between the sites with highest *Biomphalaria* spp. abundance identified by CSs and expert was consistently high (~84.1%) and increased over time.

## Conclusions and recommendations

Our results demonstrate the potential of citizen science to map putative schistosomiasis transmission sites. We therefore argue that this inclusive, powerful and cost-effective approach can be more sustainable than top-down monitoring and intervention campaigns.

## Author summary

Schistosomiasis is a snail-borne disease of great public health importance. Since drug treatment does not suffice to control the disease, the World Health Organisation now recommends including snail control and community involvement. Here we demonstrate how local inhabitants of an endemic region, after training, can detect snail populations with acceptable accuracy, at significantly lower cost. This citizen science approach could therefore generate unprecedented datasets in terms of spatiotemporal resolution and coverage. Moreover, it empowers communities and increases knowledge on snail-borne diseases and their control and prevention. We therefore argue that this community-based approach presents a valuable and sustainable compliment to classical surveillance programs, especially in remote areas, thereby generating the much-anticipated data and community support for targeted snail control.

## Introduction

Snail-borne parasitic diseases form a major public and veterinary health burden, especially in sub-Saharan Africa (SSA) [1]. The highly prevalent diseases are schistosomiasis [2] and fasciolosis [3], caused by *Schistosoma* and *Fasciola* worms, respectively. These parasites rely on freshwater snails of the genera *Biomphalaria* and *Bulinus* (for *Schistosoma*), and *Radix* (for *Fasciola*) to complete their life cycle [4–6]. In its chronic form, schistosomiasis can lead to severe health issues such as hepatic fibrosis, bladder cancer, infertility and/or stunted growth, and reduced learning ability in children [7]. Schistosomiasis has a global burden of approximately 1.43 million disability-adjusted life years [8]. In Africa, fasciolosis (liver fluke disease) predominantly affects livestock and it has been estimated to cause annual losses worth billions of dollars [9]. For instance, in Uganda, bovine fasciolosis prevalence in Kampala abattoirs is as high as 84% causing an estimated annual loss of US$ 92.4 million due to condemned liver alone [10].

Currently, the primary approach to controlling schistosomiasis is mass drug administration (MDA) among school-aged children [11]. Despite a recent modelling study using survey data from 44 countries in SSA indicating a considerable decrease in overall prevalence in the last decades [11,12], schistosomiasis continues to (re-)emerge and experts agree that MDA alone is not sufficient [13–16]. Sokolow and colleagues evaluated control programs in 83 countries/territories and estimated that interventions for schistosomiasis that integrate snail control result in a 92% reduction in prevalence, compared to a 37% reduction in programs with no or

limited snail control [17]. Simulation studies also showed an additional reduction in total disability of 40% when snail control is added to MDA [16]. In light of these findings, the World Health Organization (WHO) now advises complementing drug treatment with snail control and community involvement in the WHO roadmap for schistosomiasis elimination by 2030 [18]. Additionally, snail control will also impact animal diseases such as bovine schistosomiasis, amphistomiasis and fasciolosis, that are transmitted by the same or sympatric snail species, aligning with the One Health approach and steering away from a single-disease focus.

To minimize the environmental impact of snail control, a focal or targeted approach is necessary, whether it involves physical, chemical or biological methods [13,14,19]. However, to design effective snail control interventions, we need detailed information on snail population dynamics at fine spatial and temporal scales, and a thorough understanding of the biotic and abiotic factors that influence these dynamics [14]. Unfortunately, there is currently a severe shortage of trained malacologists worldwide, particularly in SSA, where schistosomiasis is most prevalent [14,20].

One promising method to scale up data collection is to develop a citizen science-centred approach [21], where members of the general public are involved in scientific efforts within their communities [22,23]. Citizen science involves collecting large volumes of research data, or conducting scientific experiments that are often in response to the local community or societal needs. While citizen science is still in its early stages in Africa [21,24], and never been applied in schistosomiasis control, it has recently been proven to be a viable alternative or complement to monitor vectors of malaria, Lyme disease, yellow fever, and Zika [25–29].

While the quality and reliability of citizen-generated data is often questioned [30–32], assessing data quality in citizen science projects is not straightforward due to the project-dependent nature and amount of incoming data, as well as the variability in required protocols for data quality [31]. Successful projects often deploy a combination of strategies including iterative designs, methods standardization, systematic capture, classification and data-entry procedures, refresher training for citizens, ongoing feedback, registration of metadata, statistical analysis to detect and correct potential bias, and expert validation [31,33]. However, implementing such tailored measures to monitor and evaluate data quality can be limited by associated costs in terms of human and material resources [32,34,35]. Nonetheless, thorough validation processes are necessary to harness the potential of citizen-generated data for advancing scientific knowledge and guiding evidence-based policy [36,37].

In this study, we analyse data obtained in the project 'Action Towards Reducing Aquatic snail-borne Parasitic diseases' (ATRAP), which established a citizen scientist network in Uganda consisting of 25 citizens that monitored snail populations at fixed sites. To assess the quality of the collected data, the same sites were also sampled by an expert malacologist. Our overall aim was to explore the potential of citizen science in mapping putative transmission sites for guiding targeted snail control by assessing the extent and type of observational biases. Specifically, we compared both datasets and assessed the extent of agreement in (i) snail occurrence (presence/absence)—referred to as binary agreement, and (ii) snail abundance—evaluated as consistency and numerical agreement.

## Materials and methods

### Ethical considerations

This study was approved by the Mbarara University of Science and Technology Ethical Review Committee with reference number MUREC 1/7, and by the Uganda National Council of Science and Technology under the reference number NS148ES. In addition, written informed consent was sought from the citizen scientists who participated in this study. The extent of

disclosure of the information the CSs provided, and their facilitation were fully discussed and formalised in a memorandum of understanding before the commencement of the data collection process. The memorandum of understanding was renewed annually following a refresher training to allow the CSs to are their experiences and expectations.

The study was conducted in the southern part of Lake Albert that is characterised by diverse hydrological settings–upland small streams and rivers (here categorised as stream site type), swamps and stationary pools (categorised as wetland), artificial spring wells (categorised as spring), and lacustrine shorelines and sheltered bays (categorised as lake). The sites are frequently used by the local communities increasing the risk of snail-borne disease infections. See S1 Fig for a detailed description of sampling site types. Briefly, field exploration visits were conducted to identify candidate water contact sites for the monitoring programme in the endemic Lake Albert region. Topographic gradients (lake or upland), hydrological gradients (small streams, large rivers, artificial springs, wetlands, lake) and human water contact intensity (very low to very high) were considered. Community knowledge about schistosomiasis, implied from presence of a routine MDA programme, guided the selection of the study sub counties, to motivate the participants to intervene in a known problem to them. This resulted in 77 study water contact sites in the selected eight sub counties in Kagadi and Ntoroko districts. A water contact site was defined as a delineated part of a water body for human activities such as fishing, bathing, swimming etc., for at least part of the year [38]. For consistence and comparison, a 10-metre stretch was delineated at each site while sites smaller than approximately 100 m$^2$ were considered entirely. When Lake Albert experienced flooding and recession of the water levels during this study period, the new sample sampling locations (15 affected sites) were agreed upon between the expert and respective citizen scientists.

The citizen scientists (hereafter referred to as CSs) were recruited based on recommendations from community leaders in collaboration with the ATRAP team [38]. Apart from a mobile phone, a backpack and t-shirt with the ATRAP logo, the CSs also received monthly financial compensation to cover transportation costs and data bundles [38]. The established network was operational from March 2020 until February 2023 and each CS collected data on a weekly basis at the designated water contact sites (2–4 per CS) in their village. The 25 CSs were trained in snail sampling and identification to genus level, focusing on species belonging to *Biomphalaria*, *Bulinus* and/or *Radix*. The procedures for site selection, training of CSs, snail sampling, data submission via the Kobo collect mobile application [39], and validation were described in detail by Brees et al. [38]. Furthermore, a WhatsApp group was created, facilitating communication between citizen scientists and experts. This platform streamlined interaction and enhanced collaboration. Moreover, the refresher trainings and awareness campaigns annually organised contributed to the citizens motivation. In addition, direct communication lines were open through phone calls with the project administrative assistant and the research team members. This enabled quick problem-solving whenever technical (e.g. with the smartphones) and personal project-related challenges arose.

The reporting protocol consisted of multiple-choice and open questions about the presence, identity, and quantity of snails, GPS location, sampling duration, and time (S1 Text). At each mapped water contact site, sampling occurred within a diameter of 10 m. The CSs, wearing appropriate protective gear (latex gloves and gumboots), actively searched for snails for ca. 30 minutes using a hand-held scoop net attached to a two-meter-long metallic handle. The collected snails were sorted into groups (genus level) based on their morphological features as learned during the CSs' training. *Biomphalaria*, *Bulinus* and *Radix* snails were counted and recorded in the questionnaire, while snails from other genera were classified as 'pool'. Subsequently, all the snail groups were placed on a graph paper and photographed at close range. The picture was then uploaded to a central server for later verification. All the collected snails

were returned to the site after recording the information to minimise the overall environmental impact as well as the snail population dynamics in particularly, which we aimed to detect through citizen science sampling.

The verification protocol was designed to minimise *reporting errors*, i.e. the difference between sampled data by the CSs and the data submitted in the questionnaire [38]. Reporting errors minimization was done by using skip logic and favouring multiple-choice questions, setting up a robust semi-automatic validation protocol that flags faulty entries, providing regular feedback, and organising an annual refresher training for the CSs. The data validation code developed in Python is available on GitHub at https://github.com/CiSciUganda/data_validation/blob/main/Automatic_KOBO. See Brees et al. [38] for a detailed description of semi-automatic validation.

Next to the automatic verifications, manual verifications were done by an experienced researcher (NVB) based on the submitted pictures. The pictures of the remaining snail genera labelled as 'pool' were used to verify any false absence. Any snail misclassification, inaccurate sampling location, and incorrect scoop time (correct scoop time = 30 ± 10 minutes) were identified and flagged as reporting errors. Each of the snail genera (*Biomphalaria*, *Bulinus* and *Radix*) was independently verified since reporting errors for one genus do not affect the validity of correct reports for another genus. All reports with reporting errors were filtered out (<2.5% of total reports) before further analysis.

On the other hand, *observational errors* relate to the difference between the snail presence and abundance reported by the citizens, and the (actual) *in situ* situations. This bias could not be tested by Brees et al. [38] due to a lack of 'ground-truth' data. Therefore, a trained malacologist (JT) (hereafter referred to as the 'expert') visited monthly the same sites that were monitored by the CSs weekly, following the same standardised snail collection protocol. The 'expert' dataset was used as a proxy for the 'ground truth', allowing a thorough comparison with the CSs data. Each CS sampled two to four water contact sites, but three of these sites were excluded from our analysis as the expert was unable to regularly access these remote sites. As a result, 73 out of 76 sites monitored by 24 of 25 CSs on a weekly basis were considered for data analysis. We included data from June 2020 to April 2022, when the expert stopped the monthly sampling campaigns. Due to the flooding of Lake Albert at the end of 2019 through 2021, the initially selected sampling locations (14 lake sites) shifted in space. Therefore, the CSs and expert agreed on new locations to monitor each time the water levels changed.

The expert identified all snails, including the 'pool' to species level using morphological identification described by Brown [40] and Mandahl-Barth [41]. The snails collected by the expert were kept for shedding experiments and for molecular analysis to identify snail infection, and for species identification. However, in the last four months of snail sampling, all snails were returned to the site after enumeration to assess the bias potentially introduced by snail removal on agreement/disagreement between the raters. The slight deviation of the expert from the CS protocol and the possible consequences for data comparison are discussed in S2 and S3 Figs showing no major source of divergence.

## Statistical analysis

**Snail presence/absence data.**   The data points collected by the CSs and the expert were matched in space and time, considering the closest observation within a period of seven days. The closest time points were selected for the pairwise comparison to minimise natural differences due to mortality, birth, and passive or active dispersal of snails between both time points [42].

Subsequently, the reported snail abundance was simplified to occupancy records (presence/absence) and compared in terms of binary agreement or disagreement. The extent of

agreement between the CSs and expert data was expressed as a percentage of all paired data across the site types. Site type was used as the grouping variable as the categories (lake, spring, stream or wetland) have distinctive characteristics. Agreement/disagreement was split into four components of the confusion matrix. Agreement consisted of true positives (TP) when both the expert and CSs independently recorded snail presence at a site and true negatives (TN) when both reported snail absence at a site. Disagreement was split into false negatives (FN) when the expert recorded snail presence at a site while the CS did not, and false positives (FP) when the CS recorded snail presence at a site, but the expert did not. False positives and false negatives in this case are thus considered relative to the expert data (the so-called ground truth). As such, 'false' observations in our case can thus either be caused by observation errors by CSs, the expert, or actual differences in occurrence due to changes in the local conditions (paired samples can be as much as seven days apart).

Sensitivity describes how well an observer detects the presence of objects/phenomena when they are actually present [43]. It is calculated by dividing the TP by the sum of TP and FN and expressed as a percentage. Expert-collected snail data was used as the reference for the computation of sensitivity of CSs in detecting snail presence at a site. A statistical analysis was then performed to further investigate why the expert and CS data might differ. The variables considered for this are summarised in S1 Table. A generalised binomial logistic mixed-effects regression (GLMER) was used to analyse the binary agreement of the expert with the CS snail presence/absence data. GLMER treats data points as grouped (hierarchical) considering the nested nature of data based on a random effect such as repeated measurements of the same subject [44]. Our dataset consisted of repeated observations of sampling locations that are nested at the CS level (S4 Fig). Site type, CS ID, the difference in sampling dates between CS and expert, and snail abundance were tested as predictors (fixed effects) of binary agreement between the expert and CS snail presence/absence data (S4 Fig). Sampling location (site) nested between CS ID was considered as a random effect and included as a random intercept to avoid autocorrelation (repeated observations at the same site and/or by the same CS). The snail abundance reported by the expert was transformed to a count per 30 minutes and used as an independent predictor of binary agreement in presence/absence of snails at a site.

A generalised linear mixed model framework of the GLMER is described following Ugwu and Zewotir [45]. Here $Y_{ij}$ is the binary response (agree/disagree) in the presence of snails between the CS at the $j$th location and the expert, with $P(Y_{ij} = 1)$ denoting the probability that at an $i$th time and the $j$th site, the CS agrees in snail occupancy with the expert. The GLMER then takes the following form:

$$P\left(Y_{ij} = 1\right) = \frac{exp(\eta_{ij})}{1 + exp(\eta_{ij})} \tag{1}$$

With

$$\eta_{ij} = \beta_0 + \beta_1.ID_j + \beta_2.ST_j + \beta_3.DD_{ij} + \beta_4.NS_{ij} + \gamma.(1|ID_j/Site_j) \tag{2}$$

Where, $ST_j$ is the site type, $DD_{ij}$ is the sampling date difference, $NS_{ij}$ is number of snails for each genus, $ID_j$ is the citizen scientist identification number at $Site_j$ and $Site_j$ is the sampling location. $\beta$ denotes the fixed effects coefficients while $\gamma$ denotes random effect coefficients.

A simplified model for each of the genera was developed by a manual stepwise step-up forward inclusion procedure. The inclusion of independent variables was based on a minimization of the Akaike information criterion (AIC) to balance model complexity and explanatory power [44]. The model was explored for collinearity through the calculation of the generalised variance inflation factor (VIF) values. The VIF of three was considered the threshold for

collinearity [46]. The reduction in model deviance (chi-square) was used to evaluate improvement in model fit after inclusion of independent variables compared to the null model. The function *somers2()* was used to compute Somers' $D_{xy}$ and the C index of concordance [47]. Somers' $D_{xy}$ is a rank correlation between predicted probabilities and observed responses and takes values ranging from 0 (randomness) and 1 (perfect prediction), with values greater than 0.5 considered meaningful. On the other hand, C is an index of concordance between the predicted probability and observed response. Values of C of 0.5 indicate randomness, 0.8 indicate the model has real predictive capacity while one (1) indicates perfect prediction [47]. All the analyses were done in R statistical software (v4.1.0; R Core Team 2021).

**Snail abundance data.** An exploratory analysis was performed to visualize the concordance between the snail abundance reported by the citizens and the one reported by the expert through time; as well as relative to the site types. After visualisation, the quantitative similarity-*raters' reliability*- between reported snail abundance was measured. For this analysis, only paired observations (previously adjusted to a scoop time of 30 minutes) where both the CSs and the expert reported snail presence (i.e. the TP pairs) were considered. We assess how well snail abundance is reflected by the CSs when both expert and CS detect the respective snail genus. For a detailed measure of these similarities, the data were analysed in subsets considering the site type, and the concept of reliability subdivided into consistency and numerical agreement.

Consistency is here defined as the resemblance between rankings of abundance by both rater types (CS and the expert). To compare ordinal association between two ranks, Kendall's Tau-b correlation coefficient was used [48,49]. Kendall's Tau-b allows assessing how both rater types agree in the determination of abundance ranking of the sites. Particularly, Kendall's Tau-b coefficient is used when the analysed data presents ties within a group (observations when a certain group has two or more instances with the same quantity of snails in their sampling). Kendall's Tau-b coefficient was calculated from the following formula [48,49]:

$$\hat{t}_b = \frac{C - D}{\sqrt{\left[\frac{n(n-1)}{2} - T_x\right]\left[\frac{n(n-1)}{2} - T_y\right]}} \tag{3}$$

$$T_x = \sum_{p=1}^{ex} \frac{n_p(n_p - 1)}{2} \quad T_y = \sum_{q=1}^{cs} \frac{n_q(n_q - 1)}{2} \tag{4}$$

where C represents the concordant pairs, defined as those where the sign of $(X_p-X_q)$ equals the sign of $(Y_p-Y_q)$ whereby $X_p$ and $X_q$ are ranks of two observations by the expert and $Y_p$ and $Y_q$ are the paired observations made by the CSs and D the discordant ones where the sign of $(X_p-X_q)$ is opposite to the sign of $(Y_p-Y_q)$ [49]; n represents the number of paired observations, $T_x$ and $T_y$ represent the number of tied observations as defined above for either the CS data or the expert data, respectively; and $n_p$, $n_q$ represents observations of snail abundance at sites where expert and CS sampled within less than one week of each other. Kendall's Tau-b coefficient range from -1 to 1, with 0 indicating no correlation and (+ or -) 1 indicating a perfect correlation. For this analysis, each TP paired observation was used to assess the overall agreement in rank, without considering the nested structure of the data.

In contrast to the binary agreement defined above, the numerical agreement was defined in terms of the algebraic differences between the values reported by the raters relative to an absolute zero point; and calculated through Krippendorff's alpha coefficient. Krippendorff's alpha measures numerical agreement while correcting for chance agreement and is calculated from

the following formula [50]:

$$\alpha = 1 - \frac{D_o}{D_e} \tag{5}$$

whereby $D_o$ represents the observed numerical disagreement and $D_e$ represents the numerical disagreement that is attributable to chance. Both measures rely on values specific to frequencies reported in coincidence matrices for which the construction is specific to the data type, which in our case is ratio metric differences. For a full description of the computation of $\alpha$, we refer to Krippendorff [50]. Algebraically, when observers have a perfect match (no disagreement) $\alpha$ is equal to 1; but Krippendorff's $\alpha$ coefficient usually ranges from -1 to 1, with -1 being opposite numerical agreement and 0 no numerical agreement or numerical agreement due to chance [50].

Additionally, we analysed the factors that could influence differences between the snail abundance reported by CSs and the expert. For this purpose, a linear mixed-effect model (GLMMs) was computed considering a nested structure due to the multiple observations from the same locations made by each CS (S4 Fig). We used linear mixed-effects regression to explain the difference of the snail abundance reported between CSs and expert as follows:

$$\Delta A_{ij} = \beta_0 + \beta_1 . DD_{ij} + \beta_2 . ST_j + \beta_3 . date_{ij} + \gamma_2 . (1 | ID_j / site_j) \tag{6}$$

The counting difference between reported snail abundance (TP: true positives, cases when the CSs and the expert reported snails) for observation $i$ per location $j$, $\Delta A_{ij}$, has a normal distribution with mean $\mu_{ij}$ which is modeled as a linear function of the predictors ($DD_{ij}$: time difference in days between sampling made by the expert and CSs; $ST_j$: site type; and $date_{ij}$: scaled date of sampling) as well as a random intercept term for each $ID_j$: CS ID and $site_j$ sampling location. The scaled data represents the number of days since the first day of sampling. The collinearity of the predictors was evaluated through VIF values with a threshold of three [46]. The data of each snail genus was analysed per group: (a) data whereby snail abundance reported by expert is larger than snail abundance reported by CSs, and (b) data with lower abundance reported by the expert compared to the CSs, as we aimed to explore which factors could underly both situations.

## Results

The expert collected a total of 1,382 snail and ancillary data reports, while the CSs collected 5,398 reports until 30/06/2022. From all the reports by the CSs, snail misidentification was less than 0.02% for all genera, consisting of 111 reports on *Bulinus*, 80 on *Radix*, and 41 on *Biomphalaria*. After filtering for differences in sampling dates, errors in snail identification, and sampling time, the datasets from the expert malacologist and CSs were merged. This resulted in a total of 907 paired reports on *Biomphalaria* spp., 892 on *Bulinus* spp. and 898 on *Radix* sp. were obtained. S5 Fig illustrates the total abundance of the three snail genera collected by the expert over a 20-month period. It is worth noting that out of the 73 sites observed, 72 sites (98.6%) contained at least one of the three snail genera during the observed period.

### Presence and absence

*Biomphalaria* snails were the most abundant and widely distributed among the three genera, being recorded at 68 (93%) sites (see S5 Fig). In approximately 76% of the paired reports on *Biomphalaria*, there was a binary agreement in snail occupancy at a site (presence/absence) between the expert and CSs. However, the frequency of agreement varied across different site types, as depicted in Fig 1A. The highest agreement was observed at wetland sites (86%),

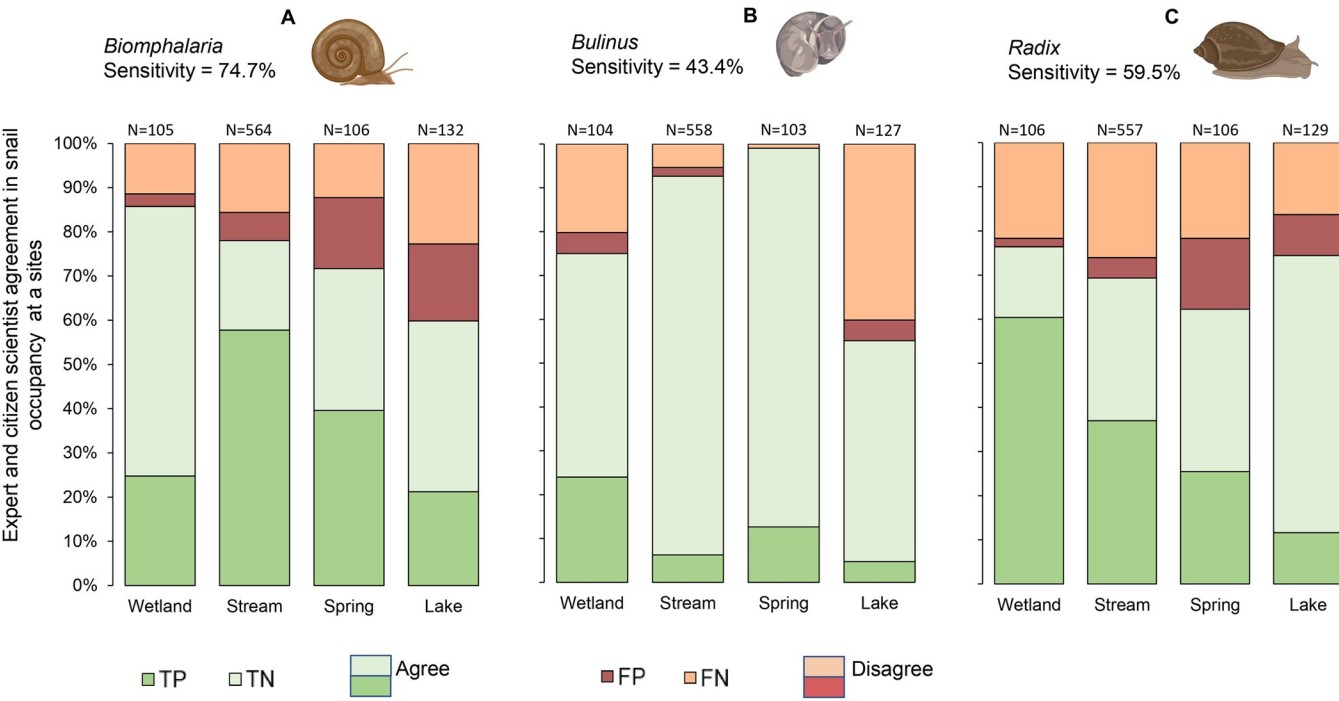

**Fig 1.** Agreement between the expert and citizen scientists collected data on the presence/absence of *Biomphalaria* (A), *Bulinus* (B) and *Radix* (C) snails at different site types. [Snail icons created with BioRender.com].

followed by streams (78%), and spring wells (72%), while lake sites had the lowest agreement (60%; N = 907). Overall, the true positive rate (sensitivity) was determined to be 74.6%.

*Bulinus* snails exhibited lower abundance and a patchy distribution within the study area. Approximately 86% of the paired reports on *Bulinus* showed binary agreement between the CSs and the expert. However, the sensitivity of *Bulinus* snail detection by the CSs was found to be low (43.4%; Fig 1B). Conversely, the *Radix* group displayed the lowest binary agreement (70%) but with a moderate sensitivity (59.5%; see Fig 1C).

The GLMER model for *Biomphalaria* was defined as $P(Y = 1) \sim SN_{i,j} + ST_j + DD_{i,j} + 1 \mid ID_j/Site_j$ (refer to Fig 2A–2C), with a probability of binary agreement, P(Y = 1), equal to 0.76. The fitted model indicated a significant prediction of agreement by the independent variables (C = 0.795, Somers' $D_{xy}$ = 0.590; N = 907), as evidenced by the reduction in deviance compared to the random effects model (chi-square (5) = 35.5; $p < 0.01$).

In general, the likelihood of binary agreement showed a significant increase as the abundance of *Biomphalaria* snails increased (odds ratio [OR] = 1.02; $p < 0.001$), particularly in wetland sites (OR = 4.30; $p = 0.002$) or stream sites (OR = 2.33, p = 0.008) (see Fig 2A and 2B). However, the binary agreement significantly decreased with an increase in the difference in sampling dates between the expert and the CSs (OR = 0.91; $p = 0.02$; Fig 2C), indicating that disagreement is at least partially driven by differences in sampling date, rather than errors committed by the sampler(s).

The order in which the expert and CSs conducted the sampling (whether the expert sampled before, after, or on the same day as the CS) was not a significant predictor of binary agreement for all three genera ($p > 0.05$), suggesting that the expert's snail removal had minimal impact on snail presence/absence (S2 and S3 Figs). The reduction in model deviance (chi-square) when including sampling order was small, ranging from 2.1 to 2.7 and statistically

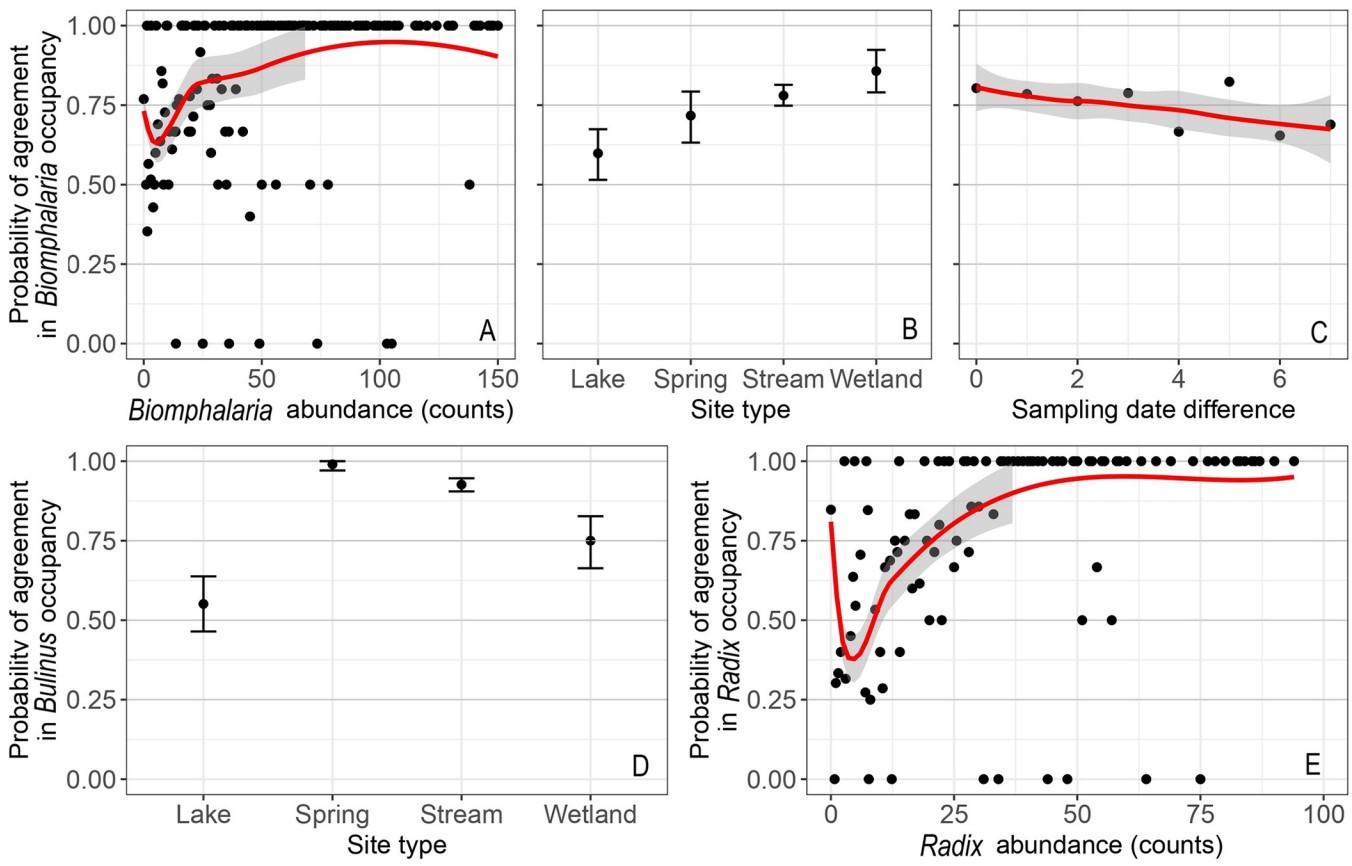

**Fig 2.** Variables influencing observed mean probabilities of agreement in snail presence/absence: *Biomphalaria* spp. presence/absence by snail abundance (A), site type (B) and sampling date difference (C); in *Bulinus* spp. presence/absence by site type (D); in *Radix* presence/absence by snail abundance (E). The 95% confidence level is indicated by the grey limits on the fitted red line for continuous data and by the error bars for site type.

insignificant. Furthermore, the effect of snail replacement by the expert in the last four months of snail sampling on the binary agreement between CS and expert data could not conclusively be determined. The agreement decreased for *Biomphalaria*, remained the same for *Bulinus* and increased for *Radix* (S2 and S3 Figs). Detailed information on the predictors of agreement in the presence/absence of *Biomphalaria* spp., including effect sizes, confidence intervals (CIs) and ORs, can be found in S2 Table.

When fitting a GLMER model for agreement in *Bulinus* spp. presence/absence, we found that only site type significantly explained the agreement (Fig 2D), with a probability, P(Y = 1), equal to 0.86. Spring sites (OR = 104.7; $p$ <0.001), stream sites (OR = 11.8; $p$ <0.001) and wetland sites (OR = 3.4; p = 0.032) exhibit a higher likelihood of achieving binary agreement compared to lake sites (Fig 2D). The fitted model demonstrated a significant improvement over the reference random effects model when site type was included as a fixed variable (chi-square (3) = 42.3; $p$ < 0.01; C = 0.889; $D_{xy}$ = 0.778; N = 892).

For *Radix*, the agreement in presence/absence was determined by the formula: $P(Y = 1) \sim SN_{i,j} + 1 \mid ID_j/Site_j$, with a probability of binary agreement, P(Y = 1), equal to 0.70. The probability of agreement between the CSs and the expert increased significantly with an increase in *Radix* sp. abundance (Fig 2E; OR = 1.01; $p$ = 0.031). Despite the simplicity of the model with only one fixed variable, the explanatory power of snail abundance was relatively high (C = 0.780; $D_{xy}$ = 0.560; N = 867). Other factors such as site type and the difference in sampling

dates between the expert and the CSs did not significantly explain the binary agreement in detecting *Radix* snail presence/absence at a site.

## Snail abundance

The total number of true positives cases (TP) were 422 for *Biomphalaria* spp., 79 for *Bulinus* spp., and 313 for *Radix*. A visual analysis of the data (Fig 3A) reveals that *Biomphalaria* snails exhibited the highest accumulated abundance, while *Bulinus* snails had the lowest abundance. Furthermore, the expert consistently reported tended higher abundance compared to the CSs for all snail genera. However, there appears to be a general agreement between the CSs and the expert regarding temporal trends, with minor variations observed (Fig 3A). Additionally, when comparing the snail abundance data between CSs and the expert, the relative dominance of each genus by site type remained consistent (Fig 3B).

To visually assess the agreement in rankings across time and space, Fig 3C illustrates the overlap in the top 15 sites (ranked by abundance) for *Biomphalaria* spp. in all comparisons of expert and CS data (TP, TN, FP, FN). All months but one exhibit more than 75% of concordance, with an average concordance of 84.1%. Additionally, the refresher training conducted in January 2021 (Fig 3D) resulted in significantly higher agreement in the subsequent months (t = -2.59; *p* = 0.023; df = 12.739).

Regarding the assessment of raters' reliability (Table 1), the consistency results (Kendall's Tau-b) displayed a constantly significant correlation coefficient for stream and wetland sites across all three genera. Among the snail genera, *Bulinus* presented the highest values of Kendall's Tau-b, implying strong relationships between the ranking of its abundance reported by the expert and the CSs at each site type, particularly at lake and stream sites. However, it should be noted that there were limited TP observations around the lake (n = 6). *Biomphalaria* and *Radix* snail abundance had the highest values of consistency among raters at wetland sites (0.541 and 0.381, respectively). For *Bulinus*, the lake site type had the highest Kendall's Tau-b but was not considered due to the small sample size (n = 6). The second-highest consistency for *Bulinus* corresponded to the stream site type where snails from this genus are more abundant (0.637).

In terms of numerical agreement (Krippendorff's α), positive coefficients were observed for spring, stream and wetland site types, indicating a positive relationship between the snail abundance reported by CSs and the expert. The stream site type exhibited the highest numerical agreement for *Bulinus* spp. (0.612), while the wetland site type showed the highest numerical agreement for *Biomphalaria* spp. (0.457). Although the values of the numerical agreement are lower than the ones of consistency, this outcome was expected since the numerical agreement metric is stricter than the one of consistency.

In our analysis, we examined the factors influencing the disparity between snail abundance reported by the expert and the CSs, and we did not identify any variables that reached statistical significance (*p*<0.05). For more detailed information about this, please refer to S3 and S4 Tables.

## Discussion

This study aimed to assess the suitability of a citizen science approach for monitoring snail intermediate hosts, with the goal of guiding targeted snail control efforts and to support species distribution modelling. In order to evaluate the quality of the collected data in relation to these objectives, we employed the concept of 'fitness for use' as suggested by Parrish et al. [25]. To ensure the generation of relevant and high-quality data, we purposively selected participants and developed a rigorous sampling protocol with fixed sampling times and sites [38]. This

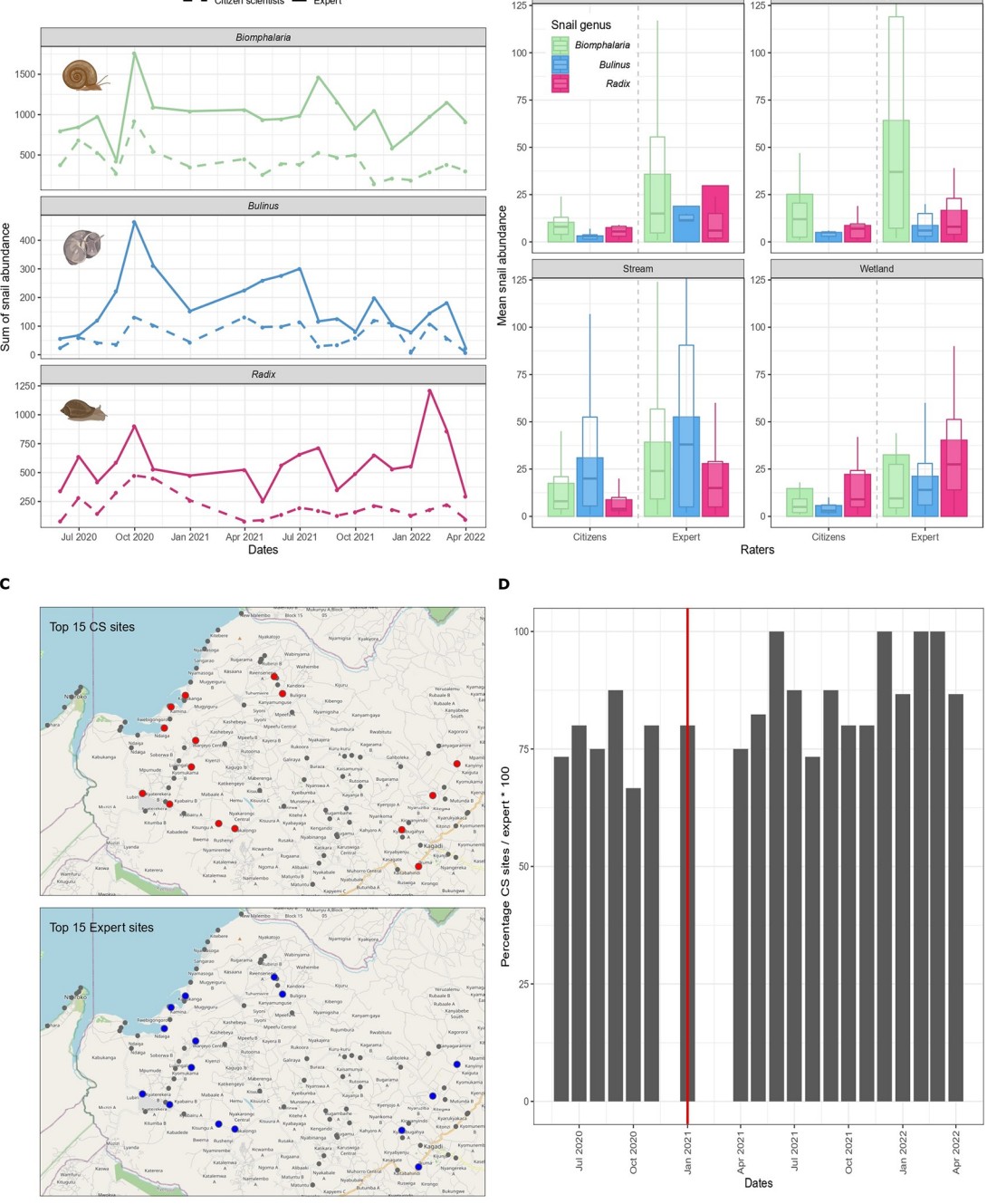

**Fig 3. Spatial and temporal trends comparison of snail abundance reported by citizen scientists and expert.** Sum of snail abundance over time per type of rater [Snail icons created with BioRender.com] (A). Mean snail abundance per site type and type of rater (limit of y-axis has been set to 120) (B). For those instances where paired reports are available for more than 40 sites, we consider the agreement in the top 15 locations in terms of abundance. Top 15 locations with the highest abundance of *Biomphalaria* spp. in February 2022 (citizen reports on the top, expert on the bottom, results from the other months showed similar trends) [OpenStreetMap database: https://www.openstreetmap.org/#map=13/1.0212/30.5998] (C). Percentage of concordance between top 15 sites of *Biomphalaria* spp. abundance reported by raters over time, the red line indicates the refresher training organised (D).

**Table 1. Raters' reliability assessment: consistency and agreement of snail abundance reported by citizens scientists and expert.**

| Site | *Biomphalaria* spp. | | | | *Bulinus* spp. | | | | *Radix* sp. | | | |
|---|---|---|---|---|---|---|---|---|---|---|---|---|
| | N | Cons | | Agree | N | Cons | | Agree | N | Cons | | Agree |
| | | K. Tau-b | p | K. α | | K. Tau-b | p | K. α | | K. Tau-b | p | K. α |
| Lake | 28 | -0.15 | 0.27 | -0.10 | 6 | 0.74 | 0.05 | -0.12 | 16 | 0.23 | 0.28 | 0.18 |
| Spring | 42 | 0.41 | < .01 | 0.27 | 13 | 0.34 | 0.12 | 0.27 | 27 | 0.11 | 0.45 | 0.06 |
| Stream | 326 | 0.38 | < .01 | 0.19 | 35 | 0.64 | < .01 | 0.61 | 206 | 0.30 | < .01 | 0.08 |
| Wetland | 26 | 0.54 | < .01 | 0.46 | 25 | 0.49 | < .01 | 0.06 | 64 | 0.38 | < .01 | 0.25 |

N is the sample size; Cons is consistency; K. Tau-b is Kendall's Tau-b; Agree is agreement and K. α is Krippendorff's alpha. Grey cells were discarded due to a small sample size.

deliberate approach helps to mitigate issues related to sampling bias, which are commonly encountered in many citizen science projects that rely on opportunistic participation and self-selection by volunteers, as seen in projects like *Mosquito alert* and *Spider in da house* [25,51]. However, in order to assess the reliability of citizen-collected data, we chose to compare it with temporally matched data collected by an expert at the same locations, considering the expert data as the 'ground truth'. This choice, and the snail sampling method (see below), has some repercussions that we need to discuss before going into the results. 'Ground truth' is difficult to establish because freshwater snail species are difficult to sample exhaustively for several reasons. Firstly, *Biomphalaria*, *Bulinus* and *Radix* snails are relatively small, with juvenile stages measuring only a few millimetres, usually hidden in the vegetation or sediment [52]. Secondly, their dispersal is not spatially continuous but rather stratified, involving a combination of passive long-range dispersal through water currents, waterbirds, and humans, as well as active short-range movements in search of food, mates or shelter from predators, heat and strong currents [53]. Additionally, circadian vertical migration has also been observed [54]. Consequently, observational inconsistencies are likely to occur, even when the same expert samples the same site at different times of the day. It is therefore worth noting that our quality control measures might be overly stringent, as some of the observation errors like false positives/negatives could be attributed to factors unrelated to the CSs' performance. In the next paragraphs we discuss two fundamental questions for prospects involving citizen scientists in snail monitoring; 1) can CSs determine snail presence at a site? And if so, 2) can they reliably monitor snail abundance?

## Can citizens correctly monitor snail presence and absence?

Our study demonstrates a substantial level of agreement between CS-collected and expert-collected data regarding the presence or absence of snails at a given site. The binary agreement rates were highest for *Bulinus* (86%), followed by *Biomphalaria* (76%) and *Radix* (70%). These agreement values are relatively high compared to the 50–80% reported among marine divers as CSs compared to experts in recording the presence of biological taxa and litter [55], but similar to the 79% reported in the *Mosquito alert* project [25]. However, our binary agreement was influenced by factors such as snail abundance and site type. For example, presence/absence of *Biomphalaria* spp. was more likely to match between the expert and CSs at sites with higher snail abundance (see Fig 2A) such as wetland sites (86%), while it was lowest at lake sites (60%). These findings could be explained by the fact that snail population dynamics are influenced by habitat type and hydrological conditions [56]. Wetland sites in the study area are relatively stable systems with minimal chances of snail migration due to their shallow lentic nature, small size, and clearly delineated boundaries. Therefore, the chances that two

independent observers will find snails at such a site are higher compared to a larger and deeper lake system. Wave action in lakes can also alter the shoreline vegetation and thus snail communities within a few hours in response to heavy winds [57]. Disagreement was also relatively high in spring wells (see Fig 2B), which are regularly unblocked by removing mud and debris, which are substrates that support snails. Spring wells are also usually located in valleys where they are prone to runoffs from heavy rains. The probability of agreement for *Biomphalaria* occupancy decreased when the difference in the number of days between sampling by the expert and CS increased (Fig 2C). This could be explained by an increasing chance for changing weather events with time, affecting the sampling sites as described above. Previous studies have shown how heavy rains or wave action can lead to passive snail dispersal [58] and subsequently low recapture values as low as 20% after one week [52]. As none of the interaction effects for site type and sampling date difference was significant, none of the site types seem to attenuate the effect of short-term fluctuations significantly, thus corroborating our statement earlier that using expert data as ground truth might be too strict.

*Bulinus* spp. presence/absence data had the highest binary agreement between the expert and CS data. However, the detection sensitivity was the lowest (43.4%) of the three snail genera studied. Lower values of sensitivity and higher values of specificity are likely when the frequency of occurrence of the event is low [43,59,60]. In our study area, *Bulinus* spp. snails were rarely encountered and when present, in relatively low numbers. For most of the sampling months (15/20), the monthly total abundance of *Bulinus* reported by all raters was less than 250 snails across the 73 sites (see Fig 3A). Consequently, the class with true negatives (TN) was very high (77%). The CSs were, therefore, able to correctly identify sites where *Bulinus* snails did not occur, increasing the chances of binary agreement with the expert (86%). Similar trends were found in *Mosquito alert* for the comparison of mosquito trapping by experts and mosquito reporting by CSs in areas with low mosquito abundance [25].

*Radix* snail presence/absence agreement between the CSs and the expert was the lowest of the three genera and the agreement was significantly predicted by snail abundance. *Radix* snails tend to attach firmly to the substrate and are often found on substrate suspended in fast-flowing waters, unlike *Bulinus* and *Biomphalaria* snails. The large shell aperture relative to the size of the snails [61,62] could provide a large surface area for the foot to attach to the substrate firmly. We observed that sampling *Radix* snails from the water using a scoop net requires extra effort in agitating the substrate, making it more difficult compared to sampling *Biomphalaria* and *Bulinus*. This could explain why there were more false negatives for *Radix* paired reports (23.6%, N = 898). However, the degree of binary agreement in *Radix* presence/absence increased from 70% to over 80% from September 2021 to February 2022 (see S6 Fig), corresponding to a period of higher *Radix* abundance reported by the expert as shown in Fig 3A. Generally, as the abundance of *Radix* snails increased, the CSs were more likely to agree with the expert, similar to *Biomphalaria* snail data. However, the decline in snail abundance from the peak in February to April 2022 was contrasted by increasing binary agreement in the same period (S6 Fig). We attribute this to the hands-on refresher training organised in January 2021, which emphasised the snail sampling technique. This underpins the importance of training and feedback in improving CSs' performance [38], especially when high quality data is desired, and when sophisticated sampling procedures and fine-scale morphological identification of organisms are involved [63–67].

The effect of CS ID on presence/absence could not be conclusively analysed due to an increase in model complexity. There is however an apparent association between CS ID and binary agreement in the presence/absence of snails. This association could be reflected by the majority of them performing well (over 50% agreement), and a few performing excellently (>80% agreement), or poorly (<50% agreement) (see S7 Fig). Future research should

investigate the factors behind this inter-individual variation, by studying the link between motivation and data quality for example.

The agreement in presence/absence is more straightforward when both the CS and the expert do (not) find snails at a site (true positives or negatives). However, the consequences of and reasons behind disagreement (false positives or negatives) are more complex. High numbers of false negatives would result in CSs underestimating the risk of schistosomiasis infection at a site. In this study, this risk is partly offset by having frequent monitoring (weekly sampling by CSs). By aggregating weekly reports by CSs and calculating the cumulative number of snails collected per month, the number of false negatives significantly decreased (chi-square (1) = 33.9; $p < 0.01$) for all genera (S8 Fig and S5 Table). False positives on the other hand, represent cases where snail pictures uploaded by the CSs are carefully validated for correct snail identification, so the reported snail was actually present. This again illustrates the limitation of our 'ground truth', as discussed above.

## Can citizens reliably monitor snail abundance in given site/habitat types?

For all snail genera, snail abundance reported by the CSs were positively correlated with those obtained by the expert. However, the correlation values were lower than those reported in other studies about intercoder reliability [68–70]. Nevertheless, most of these studies focus on medical diagnosis, which involves fewer categories of classification, and therefore fewer opportunities for disagreement. In our study, we used Kendall's Tau-b coefficient to assess the level of consistency for a discrete variable (snail counts), with values from 0 to 282 (see S1 Table). The high number of reports included in our study (422, 79 and 313 for *Biomphalaria*, *Bulinus* and *Radix* snails, respectively) can significantly impact the estimated marginal distribution of the snail counts. This, in turn, leads to an increased number of possible ranks, resulting in a greater occurrence of discordant pairs, as discussed by Agresti [49]. Consequently, these factors could possibly contribute to lower Kendall-tau values.

Similarly, the results of numerical agreement, as indicated by Krippendorff's alpha, revealed a lower agreement between CSs and the expert compared to other studies evaluating intercoder reliability in a citizen science framework [71–74]. Again, the majority of these studies were focused on the agreement between a limited number of categories (from 2 to 6) whereas we worked with ratio data that considered the algebraic differences between the values reported by the raters. Although using abundance intervals could potentially yield higher alpha values, we opted to retain the highest level of detail in our data. Nonetheless, Berland et al. [73] reported higher values of alpha ($>0.8$) when assessing intercoder agreement among novices, intermediate analysts and expert analyst counting trees. In our study, the lower values of snail abundance agreement are most likely also partly attributed to the nature of our study subjects, freshwater snails, which, unlike trees, are not immobile (see discussion above).

In line with the reported results of the presence/absence analyses, snails from the genera *Biomphalaria* and *Radix* presented higher values of consistency and numerical agreement in wetlands (Table 1), probably due to the shallow and lentic nature of these water bodies. In contrast, the highest agreement for *Bulinus* snail abundance was observed in stream sites (Table 1), which may be attributed to specific locations where *Bulinus* snails were found, such as Kolanya and Nyamasoga. These sites are characterized by small, slow-flowing streams with confined access points, which support relatively stable populations of *Bulinus*. It is important to note that both Krippendorff's α and Kendall Tau-b are calculated with all paired observations for each genus, and thus the resulting values do not account for the temporal structure of the data. When calculating these metrics for each timestep, we observe less variations

(especially for *Bulinus* snails) in Krippendorff's α than in Kendall Tau-b which is more likely to be affected by sample size (see S9 Fig).

A plausible explanation for the differences in snail abundance between CSs and the expert is the task difficulty. Determining the presence or absence of a particular snail genus requires relatively low skill, which CSs would already acquire after the initial malacological training. However, obtaining reliable abundance data comparable to the ground-truth data (expert data) requires that, in addition to identifying the potential microhabitats of the snails at the sampling sites, a vigorous and continuous sampling for 30 minutes should follow, which demands considerable physical effort and dedication. In the latter, individual factors such as motivation or skill, could contribute to the observed variations in performance among CSs (see S7 Fig). The observed increase in agreement in the ranking of high abundance sites over time supports the hypothesis of task difficulty, indicating that CSs improve their performance with more practice and experience, but also after the refresher training that stressed the importance of correct sampling (Fig 3D).

For all evaluated snail genera, consistency values were higher than numerical agreement, suggesting that using proportional abundance (Fig 3B) can be a more effective approach than exact numerical agreement, although both metrics remain relatively low and are strongly influenced by the site type.

## Can citizens help to map putative schistosomiasis transmission sites to guide targeted snail control?

Our study provides evidence of high agreement in presence/absence and spatial congruence between the CSs and the expert in monitoring snail populations. The CSs successfully identified the top 15 sites with the highest abundance of *Biomphalaria* spp. in over 84% of the matched reports. Moreover, we reported high values of the binary agreement for *Biomphalaria* spp. (76%) across all the site types, and a low number of false negatives (15.4%) (see above). The latter is important since false negatives would lead to refraining from snail control in potential transmission sites. These findings, together with the similar trend in relative snail abundance over time generated from expert and CS data, suggest that involvement of CSs has the potential to bridge the gap in snail data availability and the lack of snail experts [14,20,75]. It could generate the data needed by the national Vector Control Division for planning targeted snail control interventions, and meet the increasing need for high-resolution data in precision mapping as the efforts shift from control to elimination of schistosomiasis [14,20]. Involving local citizens could increase the spatial coverage, facilitate access to remote areas and decrease the monitoring costs (see below). A notable example to illustrate the potential is the *Mosquito alert* project in Spain [25], which utilizes a mobile app allowing citizens to report sightings of the Asian tiger mosquito. This project demonstrated collaboration between citizens and the traditional monitoring surveys improves monitoring effectiveness (>60% new information) than when either party works in isolation (<40% new information) [25]. In our study, CSs collected high-resolution data both in terms of spatial coverage, encompassing 73 sites across an area of approximately 750km$^2$, and temporal coverage, sampling on a weekly basis for 20 months. Notably, ten of these sites were located in remote areas along the shores of Lake Albert in the highly endemic Ndaiga sub-county, making the resident CS the ideal person to regularly monitor them. Therefore, we argue that citizen-generated snail data can guide targeted snail control.

However, it is important to note that in this study many locations without active schistosomiasis transmission but with high *Biomphalaria* spp. densities were found (e.g. the sites further away from the lake). Indeed, in order for schistosomiasis transmission to occur, other

requirements like human water contact and open defecation need to be met [76,77]. Therefore, CSs also record the type and frequency of water-related activities they encounter when sampling. However, to prove actual schistosomiasis transmission, human or snail infection data is needed. Since this is outside the scope of our current citizen science approach, we recommend that citizen-driven snail monitoring should only be implemented in areas with known endemicity for schistosomiasis. In the future, one could envision the involvement of citizens in eDNA monitoring of schistosome DNA in waterbodies [78], as already done for the eDNA monitoring of birds and marine species [79,80]. One could even foresee the involvement of citizens in village-based mollusciciding as done in Cameroon [81], after snail or schistosome detection in the respective water contact sites. In absence of molluscicides, citizen scientists can assist in alternative snail control measures like manual aquatic vegetation removal, which has been proven in effective snail reduction in Senegal [82]. In this case, citizen scientists instead of satellite imagery-assisted researchers identify the sites for intervention, based on the data they collect on snail abundance and human water contact patterns.

## Benefits beyond data quality

Citizen scientists offer a unique perspective on scientific and social issues that can complement or enrich those of professional scientists [34]. This fosters mutual learning between CSs and experts, allowing for subsequent exchange of knowledge and skills within their communities. Within the ATRAP project, the same CSs are also involved in community mobilisation for contextualised awareness campaigns. We have gained valuable insights from this citizen-led awareness campaigns where the CSs designed tailored messages on schistosomiasis prevention and control upon community input [83]. This led to diverse, contextualised communication approaches specific to each community, stimulating the knowledge uptake and moving away from a one-size-fits-all approach. Furthermore, Anyolitho et al. [84] reported existing myths and misconceptions regarding schistosomiasis in our study area, suggesting that bottom-up participatory approaches led by CSs can help address and reduce these misconceptions. Additionally, the process of data collection itself, which involves identifying snails associated with schistosome transmission, raises awareness about snail-borne diseases in the area [83]. We observed that the support and demand for snail control by the CSs and their fellow community members increased during the project, and witnessed a multiplication effect by having CSs safely training their family members and friends in snail sampling and identification.

From a cost-benefit perspective, there are significant advantages to engaging CSs. It would require one expert at least 80 months to collect the data collected by the 24 CSs in 20 months. Moreover, given the size of our study area, weekly monitoring is simply not feasible for one person. For instance, monitoring the 73 sites on a weekly basis would require at least four experts, amounting to €178,640, which is about 7.6 times more expensive than working with CSs. This cost is based on the involvement of a PhD student as expert, so the total price would increase if governmental experts would be involved. The current cost for our sampling campaign is €23,475 including the initial non-recurring cost of buying equipment such as the smartphone, protective gear, and scoop net (for details, see S6 Table).

In summary, we agree with Parrish et. [67] that the citizen science approach leads to increased ownership and sustainability of interventions, while providing opportunities for citizen involvement in scientific endeavours in a cost-effective manner.

## Limitations and recommendations

The time-based sampling was preferred in this study to the more accurate yet cumbersome quadrat sampling [85], as the latter would be time consuming for volunteers, difficult to

validate remotely compared to time stamps in the former, and the increased time investment would potentially lower the motivation of the CSs [86]. Irrespective of this choice, snail sampling is challenging for several reasons as discussed before.

With respect to *Bulinus* spp., the snail intermediate hosts of urogenital schistosomiasis, we need to further assess the applicability of the citizen science approach to map their distribution. Whereas the CSs correctly identified the sites where these snails occur, the relatively low spatial distribution of *Bulinus* snails resulted in few CSs monitoring the sites where they occurred, precluding an in-depth comparison between CSs and the expert.

This study was conducted in a confined geographical area with only 25 CSs actively involved. This approach can be scaled up to other areas that are endemic for schistosomiasis. Considerations like the complexity of the tasks and the ability of the CSs to learn through feedback and experience should be carefully planned in the project design. Parrish at al. [67] suggested that for complex assignments continuous expert participation is recommended to improve the data quality, which was also shown by our own study [38]. Finally, identifying and understanding the motivations of citizen scientists will further improve the design and execution of citizen science projects to ensure mutual benefits for all involved parties.

In conclusion, we demonstrate that citizen scientist-collected data on freshwater snail intermediate hosts of *Schistosoma* and *Fasciola* parasites can be used to map putative snail-borne disease transmission sites. Citizen scientists can effectively detect snail occupancy where snails are abundant, but their effectiveness decreases in sites with low snail abundance. For studies where the exact abundance of snails is needed, we recommend expert surveys or integrating both approaches. Nevertheless, our approach is fit for the purpose of guiding targeted snail control and increasing community awareness. By transferring the knowledge about snail-borne diseases among local community members, awareness and ownership increase, which is vital for the success of control and prevention strategies. We therefore argue that this inclusive, powerful and cost-effective approach can be more sustainable than top-down monitoring and intervention campaigns.

## Supporting information

**S1 Text. The data collection protocol used by citizen scientists in the Kobo Collect mobile application.**
(PDF)

**S1 Fig. Site types used in the grouping of data collected by Citizen Scientists (CS) and the expert.** Springs are underground water sources trapped by concrete walls to direct water through a metallic pipe with continuously flowing water, accessed by the communities. Stream sites are water contact sites with flowing surface water. Lake sites on Lake Albert were highly used water contact sites within fishing villages in Ndaiga Sub County and Ntoroko Town Council, on the South-Eastern side of the lake. Wetland sites consist of flooded areas for most of the year mostly characterised by rooted aquatic vegetation.
(PDF)

**S2 Fig. The effect of expert snail removal and sampling date difference with the CS for *Biomphalaria*, *Bulinus* and *Radix* snails at the different site types.** We explored the potential impact of snail removal without replacement by the expert. It was hypothesised that if snail removal impacted the extent of agreement between the expert and CS data, agreement would be higher if the CS sampled before the expert, more so on same date. Similarly, the effect of snail removal by the expert was expected to decrease with time due to migration. Thus, "Same" stands for the expert and CS having sampled a site on the same day, "Before" when the expert

sampled before the CS and "After" when the expert sampled after the CS. If the sampling date difference was between one and three, it was considered "Close" (grey error bars) and if the difference was between four and six days, it was considered "Apart" (black error bars). Except for *Biomphalaria*, the difference in sampling days did not explain significant agreement/disagreement in snail presence/absence. Across all the genera and site types, the effect of snail removal by the expert was not significant ($X^2$ (2) = (2.3, 2.1 & 2.5) p > 0.05, for *Biomphalaria*, *Bulinus* and *Radix* respectively) when the expert sampled before the CS.
(PDF)

**S3 Fig.** Observed probability of binary agreement (presence/absence) of *Biomphalaria* (A), *Bulinus* (B) and *Radix* (C) compared in the period when the expert removed snails from a site without replacement (16 months) with when the expert replaced the snails (4 months). For the three genera, the probabilities of agreement are statistically different with p<0.05. If snail removal changes snail population dynamics, the probability of agreement was expected to be lower when there was snail removal. However, except for *Radix* snails, the probability of was higher in the period when there was snail removal, contrary to our expectation. There was no clear pattern in agreement as a result of snail removal implying no clear influence of the snail removal by the expert on agreement in snail presence/absence.
(PDF)

**S4 Fig. Structure of the nested random effect implemented in the linear mixed-effect model.** ID is the CS ID, loc is the sampling site/location.
(PDF)

**S5 Fig. Map of the study area showing total abundance (expert data 06/2020–04/2022) of the studied snail genera in each water contact site, black points indicate locations without snails.**
(PDF)

**S6 Fig. Observed probability of binary agreement with time between the citizen scientists and the expert in *Radix* sp. presence/absence with time.**
(PDF)

**S7 Fig. Observed variation in the degree of agreement of each citizen scientists with the expert in detection of presence/absence of *Biomphalaria*, *Bulinus* and *Radix* snails.** We observed variations in agreement for the same citizen scientist when sampling different site types. The citizen scientist IDs were sorted from lowest to highest mean agreement with the expert for each snail genus.
(PDF)

**S8 Fig.** Analysis including cumulative abundance reported by the citizen scientists (dotted line) and the expert (full lines) per month (left panel). The differences between the graphs (Fig 3A–main text) are due to the number of points that are being considered (1037 points for each snail genera, in contrast with 911 paired reports on Biomphalaria spp., 900 on Bulinus spp. and 905 on Radix sp. from the original analysis) because originally, we only considered points that were apart +/- 7 days. The false positives increased, but also the agreement (TP + TN) increased for Radix by around 10%.
(PDF)

**S9 Fig. Values of consistency and numerical per time step (month) for all the snail genera studied.**
(PDF)

**S1 Table. Variables used in the comparison of citizen scientists' and expert collected snail data.**
(PDF)

**S2 Table. *Biomphalaria*, *Bulinus* and *Radix* predictors of agreement in snail presence/ absence between the citizen scientists and the expert, and their metrics.** Ref. stands for reference, CI for confidence interval, and significant values are in bold.
(PDF)

**S3 Table. Generalized linear mixed models output–case A.**
(PDF)

**S4 Table. Generalized linear mixed models output–case B.**
(PDF)

**S5 Table. Analysis including cumulative abundance per month (CS) /nnAbundance.**
(PDF)

**S6 Table. Summary of cost of data collection by the citizen scientist compared to the expert.** For the expert, two scenarios are considered: when one expert samples each site once a month as in our approach, and if the expert were to sample each site four times a month like the citizen scientists in this study.
(PDF)

## Acknowledgments

We greatly appreciate the contribution of all the citizen scientists without whom this study would not have been possible: Alinda Hassan, Chotum Friday, Atanasi Marisel, Opio Isingoma, Ategeka Augustine, Bahemuka Bob, Masereka Haruna, Sebakara Fobius, Nyamahunge Imelda, Businge Zabron, Tumusiime Janet, Nsenga David, Night Marygoret, Kyaligonza Noeline, Mwesige Robert, Barisigara Gard, Tusingwire Henry, Natabi Specioza, Nakingi Rose, Bamuturaki Charles, Kamukama Josias, Nuwagaba Emmanuel, Tweheyo Julius, Bahungirehe Crezestom, Ategeka Rogers, Unimu Hadijah and Fuwarinyo Richard. Support from the Health Departments through the Vector Control Units of Kagadi and Ntoroko district local governments is acknowledged.

## Author Contributions

**Conceptualization:** Julius Tumusiime, Grace Kagoro-Rugunda, Ronald Twongyirwe, Casim Umba Tolo, Liesbet Jacobs, Tine Huyse.

**Data curation:** Julius Tumusiime, Noelia Valderrama Bhraunxs, Daisy Namirembe, Liesbet Jacobs.

**Formal analysis:** Julius Tumusiime, Noelia Valderrama Bhraunxs, Liesbet Jacobs.

**Funding acquisition:** Julius Tumusiime, Grace Kagoro-Rugunda, Casim Umba Tolo, Liesbet Jacobs, Tine Huyse.

**Investigation:** Julius Tumusiime, Noelia Valderrama Bhraunxs, Grace Kagoro-Rugunda, Daisy Namirembe, Ronald Twongyirwe, Casim Umba Tolo, Tine Huyse.

**Methodology:** Julius Tumusiime, Noelia Valderrama Bhraunxs, Daisy Namirembe, Casim Umba Tolo, Liesbet Jacobs, Tine Huyse.

**Project administration:** Daisy Namirembe, Casim Umba Tolo, Tine Huyse.

**Resources:** Casim Umba Tolo, Tine Huyse.

**Software:** Noelia Valderrama Bhraunxs.

**Supervision:** Grace Kagoro-Rugunda, Christian Albrecht, Ronald Twongyirwe, Casim Umba Tolo, Liesbet Jacobs, Tine Huyse.

**Validation:** Julius Tumusiime, Noelia Valderrama Bhraunxs, Liesbet Jacobs, Tine Huyse.

**Visualization:** Julius Tumusiime, Noelia Valderrama Bhraunxs.

**Writing – original draft:** Julius Tumusiime, Noelia Valderrama Bhraunxs.

**Writing – review & editing:** Julius Tumusiime, Noelia Valderrama Bhraunxs, Grace Kagoro-Rugunda, Daisy Namirembe, Christian Albrecht, Ronald Twongyirwe, Casim Umba Tolo, Liesbet Jacobs, Tine Huyse.

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
