## [Decision Letter · Decision Letter 0]

15 Dec 2023

Dear Mr Tumusiime,

Thank you very much for submitting your manuscript "Citizens can help to map putative transmission sites for snail-borne diseases" for consideration at PLOS Neglected Tropical Diseases. As with all papers reviewed by the journal, your manuscript was reviewed by members of the editorial board and by several independent reviewers. The reviewers appreciated the attention to an important topic. Based on the reviews, we are likely to accept this manuscript for publication, providing that you modify the manuscript according to the review recommendations. 

The reviewers were generally positive in their evaluations, but still there some relevant questions to be addressed. Please provide answers and indicate the modifications to all recommendations. Please be sure to address issues such as costs, ethics, site selection, and applicability in different contexts.

Sincerely,

Guilherme L Werneck

Academic Editor

Álvaro Acosta-Serrano

Section Editor

The reviewers were generally positive in their evaluations, but still there some relevant questions to be addressed. Please provide answers and indicate the modifications to all recommendations. Please be sure to address issues such as costs, ethics, site selection, and applicability in different contexts.

Reviewer's Responses to Questions

**Key Review Criteria Required for Acceptance?**

**Methods**

-Are the objectives of the study clearly articulated with a clear testable hypothesis stated?

-Is the study design appropriate to address the stated objectives?

-Is the population clearly described and appropriate for the hypothesis being tested?

-Is the sample size sufficient to ensure adequate power to address the hypothesis being tested?

-Were correct statistical analysis used to support conclusions?

-Are there concerns about ethical or regulatory requirements being met?

Reviewer #1: Methods:

- The objectives of the study and the hypotheses being tested are clearly outlined in the text.

- The study design and methods seem suitable for the purposes of the study.

- Ethics: There is no mention of ethics review or approval for the study. It might be possible that the Ugandan authorities may not have required it since they involved malacological data.

Reviewer #2: Yes. Details queries have been imbedded in the manuscript

Reviewer #3: -Are the objectives of the study clearly articulated with a clear testable hypothesis stated?

YES

-Is the study design appropriate to address the stated objectives?

YES

-Is the population clearly described and appropriate for the hypothesis being tested?

N/A

-Is the sample size sufficient to ensure adequate power to address the hypothesis being tested?

YES

-Were correct statistical analysis used to support conclusions?

YES

-Are there concerns about ethical or regulatory requirements being met?

NO

**Results**

-Does the analysis presented match the analysis plan?

-Are the results clearly and completely presented?

-Are the figures (Tables, Images) of sufficient quality for clarity?

Reviewer #1: - The analytical approach and results reported are considered pertinent and to a good standard. 

- Figures and tables appear suitable. They are informative and complement the text adequately.

Reviewer #2: Yes

Reviewer #3: -Does the analysis presented match the analysis plan?

YES

-Are the results clearly and completely presented?

YES

-Are the figures (Tables, Images) of sufficient quality for clarity?

YES, the integration between the figure in the main body of the paper and those in the Supplementary Information is perfecr

**Conclusions**

-Are the conclusions supported by the data presented?

-Are the limitations of analysis clearly described?

-Do the authors discuss how these data can be helpful to advance our understanding of the topic under study?

-Is public health relevance addressed?

Reviewer #1: - The conclusions are adequately supported by the data.

- There is a limitations section in the text.

Reviewer #2: The authors need to align their conclusion with the aims of the study.

Reviewer #3: -Are the conclusions supported by the data presented?

YES

-Are the limitations of analysis clearly described?

YES, excellent job in the discussion of the limitation

-Do the authors discuss how these data can be helpful to advance our understanding of the topic under study?

YES

-Is public health relevance addressed?

YES, schistosomiasis and fasciolasis are disease of great public health and veterinary importance

**Editorial and Data Presentation Modifications?**

Reviewer #1: - There is a small typo in Fig 3. The graph is divided between A,B,C and D sections. The legend only lists A-C, although the text covers all 4 areas.

- The legends for Figures 1 and 3 will require to specify the sources of images to ensure no copyrights are vulnerated.

Reviewer #2: if comments are addressed the manuscriot should be good for publication

Reviewer #3: none

**Summary and General Comments**

Reviewer #1: General recommendations:

METHODS:

- I would recommned the authors to provide a minimum of information about site selection process and operational definition of water points visited. Whilst these subjects can be consulted in another reference, it'd help the reader to have a broader understanding of the study if these key points of the paper are stated explicitly in the text. 

- The project depended on a substantive commitment by citizen scientists. It would be important to highlight that monthly transportation and mobile data costs were covered by the project (if not stated, the reader may have an incorrect interpretation of the demads on residents). It would be useful to know as well if any measures were adopted to keep volunteers' motivation high throughout the study.

- Concerning ethics, it is recommended that the authors provide a statement indicating how this subject was approached. If no ethics approval was required by Uganda's MoH or what additional institutional reviews the project went through.

DISCUSSION:

- It would be recommended expand the section "Can citizens help to map putative schistosomiasis transmission sites to guide targeted snail control?" with information more directly related to schistosomiasis. That is, addressing more directly how existing or prospective snail control project designs / approaches can be directly informed by the CS evidence, and to what extent the quality of the evidecen gathered is sufficient to that effect.

Reviewer #2: Nil

Reviewer #3: This is an excellent paper aimed at testing whether data on presence and abundance of freshwater snails of medical importance gathered through citizen science is comparable, as for quality and precision, to expert-gathered data. Data on snail presence and abundance for Biomphalaria, Bulinus and Radix spp, - obligate intermediate hosts for S. mansoni, S. haematobium and Fasciola parasites, respectively - were collected weekly by 25 trained citizen scientists at 76 sites around southern Lake Albert (Uganda) for 20 months. The quality of this data was assessed by comparing it to monthly data collected by an ‘expert’ malacologist using the same sampling protocol. Generalized binomial logistic, linear mixed-effects models and a comprehensive number of additional metrix of concordance/discordance, specificity and sensitivity were used to analyze agreement between CS- and expert-gathered data. The authors found a good agreement for – and some difference among – the three snail taxa object of this study showing a good match for presence/absence data. Snail abundance estimated by citizen scientists was systematically lower than that estimated by experts, but the temporal trends were consistent between CS data and expert gathered data. The difference in abundance was highest in the case of Radix snails, as these snails have a stronger attachment to the substrate. 

The paper is beautifully written. The statistical methods are rigorous, thorough, and comprehensive (a minor note on the use of the critizied stepwise regression, but - as I personally think - legitimately applied in this paper and, anyway, just one of the several tests used by the authors, all pointing to similar conclusions). 

Results are clearly presented. Statistically significant differences or lack thereof were clearly discussed and interpreted. Strengths and limitations of CS for detection of snails of medic ally importance were honestly presented, including also benefits extending beyond the sampling activity, most importantly in terms of community engagement.

This paper is incredibly relevant as it shows that empowering local populations and training citizen scientists is doable, cost effective and produce reliable data. This approach can and should be incorporated systematically in future field studies of the intermediate host of schistosomiasis and fascioliasis. I personally consider this study as transformational for its impact. 

Also, from the technical viewpoint, this is one of the few papers that, to my humble understastanding, requires minimum revisions and can be nearly accepted as such. Kudos to the authors for having assembled such as well conceived paper, and kudos also to the citizen scientists listed in the acknowledgment, that contributed to this important study. 

I only have minor comments, points, as reported here below. I do not need top revise the manuscript once the authors have addressed them 

Line 25: change “while” with “and”.

Line 147: Altogether, reporting errors due to snail misclassification, inaccurate sampling location, and incorrect and incorrect scoop time make less than 2.5% of total reports (line 151). Personally, I think that snail classification is very different from inaccurate sampling location, and incorrect and incorrect scoop time, as it tells us also how good are citizen scientists in classifying snails (and it looks like they are very good), I am curious to know the fraction of snail misclassifications, and whether misclassification occurred especially between Bulinus and Radix. 

Line 167-169: <<in the last in the last four months of snail sampling, all snails were returned to the site after enumeration…>>: I am curios to now whether there is knowledge of prevalence of infection in the human population, and whether snails were sampled at transmission sites or elsewhere – did the authors ponder the risk of returning potentially infected snails in the water body where human infection could potentially occur?

Line 191 and following. The authors clearly define sensitivity. Below in resultsthey use specificity, but, unless I missed, they didn’t define it in this paragraph – if this is the case, I wonder whether they might want to provide the definition year.

Line 202-203: As for the nested random effects, I suggest to make explicit reference to the figure in the supplementary information “SI Appendix Fig S4”, which is really clear.

Line 257-258: maybe I am picky but, in the formulas eq. 4, the authors use “n.p” and “n.q” (number of snails sampled by CSs and experts), where here they define plain “p” and “q” – should the definition of “n.p” and “n.q” be explicitly reported in the text? 

Line 275: consider removing “and 1 perfect numerical agreement” as you have already said so at line 272.

Fig. 2: I think the letters A, B… E to identify the five panels are missing. 

As additional minor point, RE:Line2 121-127: I looked at Brees et al. 2021, but I do not think that it is all that clear how data submission via the Kobo toolbox occurred, I was wondering whether the authors could add an appendix in the supplementary information with a slightly more expanded explanation or description of the Kobo app/interface to store the data, whether each CS typed in the data at the transmission site, etc..

PLOS authors have the option to publish the peer review history of their article (what does this mean?). If published, this will include your full peer review and any attached files.

Reviewer #1: No

Reviewer #2: No

Reviewer #3: Yes: Giulio De Leo

Figure Files:

Data Requirements:

Reproducibility:

References

---

## [Decision Letter · Decision Letter 1]

11 Mar 2024

Dear Mr Tumusiime,

We are pleased to inform you that your manuscript 'Citizens can help to map putative transmission sites for snail-borne diseases' has been provisionally accepted for publication in PLOS Neglected Tropical Diseases.

Best regards,

Guilherme L Werneck

Academic Editor

Álvaro Acosta-Serrano

Section Editor

Reviewer's Responses to Questions

**Key Review Criteria Required for Acceptance?**

**Methods**

-Are the objectives of the study clearly articulated with a clear testable hypothesis stated?

-Is the study design appropriate to address the stated objectives?

-Is the population clearly described and appropriate for the hypothesis being tested?

-Is the sample size sufficient to ensure adequate power to address the hypothesis being tested?

-Were correct statistical analysis used to support conclusions?

-Are there concerns about ethical or regulatory requirements being met?

Reviewer #2: Yes

Reviewer #3: -Are the objectives of the study clearly articulated with a clear testable hypothesis stated?

YES

-Is the study design appropriate to address the stated objectives?

YES

-Is the population clearly described and appropriate for the hypothesis being tested?

NA (yes for the snail population, the sampling methods, the dwescription of CSs and Experts did)

-Is the sample size sufficient to ensure adequate power to address the hypothesis being tested?

YES

-Were correct statistical analysis used to support conclusions?

YES

-Are there concerns about ethical or regulatory requirements being met?

NO (and, from the rebuttal to reviewers' criticism, I see that the author have been really cautious)

**Results**

-Does the analysis presented match the analysis plan?

-Are the results clearly and completely presented?

-Are the figures (Tables, Images) of sufficient quality for clarity?

Reviewer #2: The comments i raised have bee adequately addressed

Reviewer #3: -Does the analysis presented match the analysis plan?

YESA

-Are the results clearly and completely presented?

YESA

-Are the figures (Tables, Images) of sufficient quality for clarity?

YES

**Conclusions**

-Are the conclusions supported by the data presented?

-Are the limitations of analysis clearly described?

-Do the authors discuss how these data can be helpful to advance our understanding of the topic under study?

-Is public health relevance addressed?

Reviewer #2: Yes

Reviewer #3: -Are the conclusions supported by the data presented?

YES

-Are the limitations of analysis clearly described?

YES

-Do the authors discuss how these data can be helpful to advance our understanding of the topic under study?

YES

-Is public health relevance addressed?

YES

**Editorial and Data Presentation Modifications?**

Reviewer #2: (No Response)

Reviewer #3: I have no further modification to request (in my first review I mentioned that while the authors provide the definition of sensitivity now line 222, specificity is mentioned at line 494 but not defined in Methods. I do understand authors' response that they are only interested in sensitivity for this study, and I agree with it. In theory I do not see why they cannot report the definition of specificity. On the other hand, everybody in our fields knows what specificity and sensitivity are, and it is however information that can be easily retrieved on line, so I do not see this is a significant omission, and I am ok to accept the paper as such)

**Summary and General Comments**

Reviewer #2: The authors have addressed the comments i had raised.

Reviewer #3: the authors responded to all the reviewers' criticism and concern, the paper is not ready for publication, I see it as a significant contribution to understand the role of citizen science and community engagement for field malacology.

I haven't checked specifically data availability as I didn't understand where to look at on the journal web site, but the authors mentioned in their response that they made available in the online supplementary information additional data on the snail sampling protocol, the code of the application to store the data, etc. - so it seems that there is everything needed. The editorial office can easily double check it

PLOS authors have the option to publish the peer review history of their article (what does this mean?). If published, this will include your full peer review and any attached files.

Reviewer #2: **Yes: **Dr. Chester Kalinda

Reviewer #3: **Yes: **Giulio De Leo

---

## [Editor Report · Acceptance letter]

27 Mar 2024

Dear Mr Tumusiime,

We are delighted to inform you that your manuscript, " Citizens can help to map putative transmission sites for snail-borne diseases ," has been formally accepted for publication in PLOS Neglected Tropical Diseases.

Best regards,

Shaden Kamhawi

co-Editor-in-Chief

Paul Brindley

co-Editor-in-Chief
